# Muricholic Acids Promote Resistance to Hypercholesterolemia in Cholesterol-Fed Mice

**DOI:** 10.3390/ijms22137163

**Published:** 2021-07-02

**Authors:** Dany Gaillard, David Masson, Erwan Garo, Maamar Souidi, Jean-Paul Pais de Barros, Kristina Schoonjans, Jacques Grober, Philippe Besnard, Charles Thomas

**Affiliations:** 1Center for Translational Medicine, UMR1231 INSERM-uB-AgroSupDijon, Université de Bourgogne Franche-Comté (UBFC), 21000 Dijon, France; dany.gaillard@cuanschutz.edu (D.G.); david.masson@u-bourgogne.fr (D.M.); jppais@u-bourgogne.fr (J.-P.P.d.B.); jacques.grober@u-bourgogne.fr (J.G.); 2Department of Cell & Developmental Biology, and The Rocky Mountain Taste & Smell Center, University of Colorado Anschutz Medical Campus, Aurora, CO 80045, USA; 3LipSTIC LabEx, Université de Bourgogne Franche-Comté (UBFC), 21000 Dijon, France; 4Biochemistry Department, University Hospital François Mitterrand, 21000 Dijon, France; 5IGBMC, CNRS UMR 7104, INSERM U 1258, 67400 Illkirch, France; erwanga@gmail.com; 6Institut de Radioprotection et de Sûreté Nucléaire (IRSN), 92260 Fontenay-aux-Roses, France; maamar.souidi@irsn.fr; 7Lipidomic Facility, Université de Bourgogne Franche-Comté (UBFC), 21078 Dijon, France; 8Institute of Bioengineering, Life Science Faculty, École Polytechnique Fédérale de Lausanne, 1015 Lausanne, Switzerland; kristina.schoonjans@epfl.ch; 9Physiologie de la Nutrition, AgroSup Dijon, 21000 Dijon, France

**Keywords:** bile acids, cholesterol, hypercholesterolemia, LXR, FXR

## Abstract

Background and aims: Hypercholesterolemia is a major risk factor for atherosclerosis and cardiovascular diseases. Although resistant to hypercholesterolemia, the mouse is a prominent model in cardiovascular research. To assess the contribution of bile acids to this protective phenotype, we explored the impact of a 2-week-long dietary cholesterol overload on cholesterol and bile acid metabolism in mice. Methods: Bile acid, oxysterol, and cholesterol metabolism and transport were assessed by quantitative real-time PCR, western blotting, GC-MS/MS, or enzymatic assays in the liver, the gut, the kidney, as well as in the feces, the blood, and the urine. Results: Plasma triglycerides and cholesterol levels were unchanged in mice fed a cholesterol-rich diet that contained 100-fold more cholesterol than the standard diet. In the liver, oxysterol-mediated LXR activation stimulated the synthesis of bile acids and in particular increased the levels of hydrophilic muricholic acids, which in turn reduced FXR signaling, as assessed in vivo with *Fxr* reporter mice. Consequently, biliary and basolateral excretions of bile acids and cholesterol were increased, whereas portal uptake was reduced. Furthermore, we observed a reduction in intestinal and renal bile acid absorption. Conclusions: These coordinated events are mediated by increased muricholic acid levels which inhibit FXR signaling in favor of LXR and SREBP2 signaling to promote efficient fecal and urinary elimination of cholesterol and neo-synthesized bile acids. Therefore, our data suggest that enhancement of the hydrophilic bile acid pool following a cholesterol overload may contribute to the resistance to hypercholesterolemia in mice. This work paves the way for new therapeutic opportunities using hydrophilic bile acid supplementation to mitigate hypercholesterolemia.

## 1. Introduction

Hypercholesterolemia is an established risk factor for atherosclerosis and cardiovascular disorders [1,2,3,4]. Since 1960, animal models are widely used for studying the pathophysiology of hypercholesterolemia and atherosclerosis as well as for the preclinical assessment of drugs in this context [5]. Compared to other laboratory animal models (rats, hamsters, rabbits), the mouse is the most common animal model used in this field (Appendix A). Non-human primate (NHP) models are superior for investigating cholesterol (CS) metabolism and related disorders [6]. Nevertheless, cost, ethical considerations, skills, and lack of biological tools curb the use of NHP in favor of small laboratory animals. Notably, mice are highly resistant to hypercholesterolemia. Plasma cholesterol levels in mice fed a 0.5 to 2% (*w*/*w*) CS diet remain unchanged or only mildly increase up to 1.5-fold compared to mice fed a control diet [7,8,9,10,11,12]. Similar observations were made in rats [13,14]. By contrast, hamsters and rabbits develop severe hypercholesterolemia when fed lower or similar dietary CS overload for comparable durations, where plasma CS rises 3- to 5-fold [14,15,16] and 10- to 30-fold [17,18,19,20], respectively. In mice, hypercholesterolemia and atherosclerosis can be observed when genetically engineered mouse models with deficient clearance of apoB-containing lipoproteins such as Apoe-deficient mice (*Apoe*^−/−^) or Ldlr-deficient mice (*Ldlr*^−/−^) are fed a western diet (high saturated fatty acids, high CS) [5]. Alternatively, atherosclerotic lesions can be observed with dietary manipulation, such as the Paigen diet, which consists in adding cholic acid (CA) (usually 0.5% *w*/*w*) to a high fat and high CS (usually 1.25% *w*/*w*) diet [21,22]. In mice, CA is one of the main Farnesoid-X-Receptor (FXR) agonists. Therefore, dietary supplementation with CA results in the down-regulation of *Cyp7a1*, the rate-limiting enzyme in the conversion of CS into bile acids (BAs) in the liver [23].

Several mechanisms were proposed to explain the differences in CS metabolism and sensibility to dietary CS overload between mice and humans. First, plasma lipoprotein profiles are markedly different. In mice, plasma CS is predominantly localized in high density lipoproteins (HDL), while very low concentrations of low density lipoproteins (LDL) are observed [6]. This lipoprotein profile is typically associated with a lower risk of cardiovascular disease [24] and is maintained even on a high CS diet [25]. This is due in part to the absence of cholesteryl ester transport protein (CETP), a key enzyme involved in the transfer of cholesteryl ester from HDL to apoB-containing lipoproteins. CETP is absent in mice but is expressed in hamsters, rabbits, and NHP, which are hypercholesteromia prone species [6]. However, transgenic mice expressing human *CETP* also display high HDL-CS levels [26], suggesting that rapid clearance of LDL particles and processing of CS in the liver are likely key contributors to the resistance of mice to dietary CS overload.

Hepatic conversion of CS into BAs may contribute to the resistance of murine models to diet-induced hypercholesterolemia. CS homeostasis is directly linked to BA homeostasis, which is maintained via a balance between liver synthesis (primary BAs), recycling in the liver through the enterohepatic cycle, and elimination in feces and urine. BAs are synthesized from CS in the liver, excreted in the duodenum, taken up in the ileum, and recycled by the liver from the portal vein. About 5–10% of BAs are discarded in feces and urine, resulting in de novo conversion of CS into BA in the liver [27,28,29]. BA homeostasis relies on a tight regulation of liver BA synthesis pathways as well as BA transporters in the gut, the liver, and the kidney by FXR, liver-X-receptors (LXRs), and sterol-responsive element binding proteins (SREBPs) signaling [30,31,32]. Approximately 90% of the pool of CS is metabolized into BAs in the liver via two distinct pathways: the classical (or neutral) pathway and the alternative (or acidic) pathway. The classical pathway, which involves the rate-limiting enzyme CYP7A1, accounts for at least 75% of the total BA pool, as demonstrated by the ablation of Cyp7a1 in mice [33,34]. In humans, CYP7A1 produces mainly chenodeoxycholic acid (CDCA), which is the most abundant BA species with CA [35,36,37]. In mice, CDCA is converted into muricholic acids (MCAs) by the CYP2C70 enzyme in the liver [38,39]. This confers mice a specific BA profile characterized by a low hydrophobic index (HI) or Heuman index [40]. Using this method, the HI of the BA pool is 0.45 in humans and −0.09 in mice [41]. 

Since CYP7A1 is essential to the conversion of CS into BAs, a particular focus is placed on investigating its gene regulation. *Cyp7a1* expression is regulated by BA and oxysterols via nuclear receptors FXR and LXR in mice and humans [29,42,43,44,45]. The LXR pathway was proposed to be more prevalent than the FXR pathway in mice and rats [11,13,46,47], whereas the opposite was claimed in atherosclerosis-prone species such as rabbits and humans [8,48,49]. This is namely due to the absence of a functional LXRE in human *CYP7A1* gene promoter [50,51]. Nevertheless, additional species-specific regulation of liver BA synthesis may contribute to the discrete sensitivity to dietary CS overload in each species. Moreover, although fecal BA represents a major route for body CS removal, the regulation of BA transport and elimination across the body must be considered as well.

In the present study, we assessed both BA and CS metabolism in mice fed a standard diet or a CS-enriched diet (2% CS, *w*/*w*) through an integrated approach aiming to assess the coordinated regulation of BA synthesis pathways, BA transport, and CS homeostasis in the liver, the gut, and the kidney.

## 2. Results

### 2.1. Cholesterol Feeding Reduces FXR Signaling through Enhanced Hydrophilic Bile Acid Synthesis

Eight-week-old male C57BL6/J mice were fed a CS-enriched diet (2% CS, *w*/*w*) or a standard diet (0.02% CS, *w*/*w*) for 15 days. As expected, CS-fed mice did not develop hypercholesterolemia or hypertriglyceridemia (Figure 1A,B). 

In addition, food and water intake as well as body weight gain remained unchanged between groups (Appendix A and data not shown). Because hepatic conversion of CS into BA accounts for 50% to 70% of body CS elimination in mice [25], we assessed the expression and the enzymatic activity of the key enzymes of the classical (CYP7A1, CYP27A1, CYP8B1) and the alternative (CYP27A1, CYP7B1) BA synthesis pathways. *Cyp7a1* expression and enzymatic activity were enhanced in CS-fed animals (Figure 2A,B). Consistently, the level of β-MCA, which is produced by CYP2C70 from CDCA, the main product of the CYP7A1 pathway, was greatly increased in the liver (Figure 2F and Appendix A). In parallel, we observed a reduction in proportion of CA (Figure 2F and Appendix A), although the activity of CYP8B1 was significantly augmented (Figure 2B). This increased enzymatic activity of CYP8B1 was not associated with elevated mRNA levels (Figure 2A,B).

Expression levels of *Cyp27a1* and *Cyp7b1* were also unchanged (Figure 2A), while mRNA levels of the two key enzymes controlling BA conjugation, *bile acid-CoA synthase* (*Bacs*) and *bile acid-CoA:amino acid N-acetyltransferase* (*Bat*), were increased in CS-fed mice (Figure 2C). As *Cyp7a1* mRNA levels were significantly upregulated, we next monitored oxysterols, which are established LXRα ligands and master regulators of *Cyp7a1* in mice [10,42,46]. As expected, oxysterols levels were higher in the liver of mice fed the CS-enriched diet than mice fed the control diet (Figure 2D), supporting the enhanced *Cyp7a1* expression (Figure 2A). Along with increased levels of LXRα ligands, we observed a marked drop in the FXR agonists/FXR antagonists ratio in the liver (Figure 2G), whereas the total amount of BAs in the liver was increased (Figure 2E). Interestingly, the enhanced BA synthesis was biased in favor of MCAs (Figure 2F and Appendix A), which are potent FXR antagonists [52,53], indicating that, even though BA levels were increased in CS-fed mice, FXR tonus was reduced, as shown in the liver of CS-fed *Fxr*-Luc reporter mice (Figure 3A,B). In these animals, total body imaging of FXR-dependent luminescence [54] did not reveal any difference between standard diet and CS-enriched chow (Figure 3A).

Nonetheless, tissue-specific assessment of FXRE transactivation by qPCR quantification of pd-Luc expression demonstrated that FXR activation was reduced in the liver of CS-fed mice (Figure 3B). A similar trend was observed in the ileum, whereas no changes occurred in the kidney (Figure 3B). In line with these findings, expression of the prototypical FXR target gene *small-heterodimer-partner* (*Shp*) was reduced in the liver of CS-fed mice (Figure 3C). *Shp* expression trended downward in the ileum and the kidney but did not reach statistical significance (Figure 3D,E). Therefore, diversion of BA synthesis in favor of hydrophilic BAs supports the dominance of LXR over FXR signaling in the liver of CS-fed mice in addition to the continuous conversion of CS into BA without the hydrophobic BA-mediated FXR brake, as observed in rabbits or humans [29,49]. In addition, CS-dependent SREBP2 signaling was also impacted by the rise in CS levels in CS-fed mice (Figure 3C and Figure 9D). Specifically, expression of SREBP2 target genes, *Ldlr* and *Hmg-CoA reductase*, was significantly weakened in the liver (Figure 9G), and the level of the mature form of SREBP2 dropped in the ileum (Figure 3F).

### 2.2. Cholesterol Feeding Promotes Fecal and Urinary Excretion of De Novo Synthesized Hydrophilic Bile Acids

Following synthesis in the liver, BAs are excreted along with CS and phospholipids in bile canaliculi to generate mixed micelles. The fate of BAs is determined by the regulation of BA transporters in the gut, the liver, and the kidney. Therefore, we assessed the impact of a CS-enriched diet on BA transporter expression and BA fate in these three key organs.

CS feeding did not affect the mRNA and protein expression of the bile-salt export pump (BSEP, ABCB11), the main BA transporter at the canalicular side of the hepatocytes (Figure 4A,B).

This transporter mainly ensures the excretion of monovalent conjugated BA (tauro- and glyco-conjugated), as does, to a lesser extent, the multidrug resistance protein 1a (MDR1a) [27,55]. The expression of this canalicular BA transporter was unaltered here (Figure 4A). Divalent conjugated BA (particularly sulfo- or glucurono-conjugated) together with reduced glutathione or non-BA conjugated anions such as bilirubin glucuronides can be excreted via the multidrug resistance-associated protein-2 (MRP2, ABCC2) [27,56]. Although *Mrp2* mRNA levels were slightly reduced (Figure 4A), MRP2 protein levels were unchanged in CS-fed animals (Figure 4B). Altogether, these data suggest that the excretion of BAs into the bile is not affected by dietary CS overload and is sustained to limit deleterious accumulation of BAs in the liver when BA synthesis is markedly increased.

In the gut, BAs are predominantly reabsorbed in the ileum to be directed toward the liver for recycling. The apical sodium-dependent bile acid transporter (ASBT, SLC10A2) located at the apical side of the ileocytes is the major intestinal BA transporter [57,58,59]. By contrast, MRP2 may constitute an apical efflux gate [27]. Basolateral efflux of BAs into the portal circulation involves a heterodimeric complex formed by the organic solute transporters α and β (OST α, OST β) [60,61] and, to a lesser extent, MRP3 [62,63]. Consistent with our previous report [31], ASBT expression was drastically reduced at both mRNA and protein levels (Figure 5A,D). We previously demonstrated that CS-mediated regulation of *Asbt* is SREBP2-dependent [31]. Accordingly, we noticed a marked reduction of mature SREBP2 levels in the ileum of CS-fed mice (Figure 3F). Although *Mrp2* expression was unchanged (Figure 5B), we report, for the first time, a marked reduction of OSTα/β at protein levels upon high dietary CS feeding (Figure 5C,D). Notably, *Ostα*/*β* are FXR target genes [64,65]. We observed a trend for reduced FXR mRNA levels and reduced FXR activation in the ileum in CS-fed mice (Figure 3B,D). This coincides with the striking reduction in the fecal FXR agonist/FXR antagonist ratio (Figure 5G) and supports the assumption that CS-mediated deactivation of FXR results in diminished OSTα/β expression. 

This down-regulation of the main ileal BA efflux transporters at both apical and basal sides of the ileocytes suggests that fewer BA were recycled. Consistently, mice fed a 2% CS diet had increased fecal BA excretion (Figure 5E), in favor of neo-synthesized BAs, namely MCAs (Figure 5F and Appendix A). Whereas the proportion of primary BAs in the feces was enhanced (26.4 ± 1.25% in control vs. 40.7 ± 3% in 2% CS fed mice, *p* = 0.001), the fecal excretion of secondary BA deoxycholic acid (DCA) produced from CA by colonic bacteria was significantly reduced (68.6 ± 1.2% in control vs. 53.3 ± 2.6% in 2% CS fed mice, *p* = 0.0003) (Figure 5F and Appendix A). Notwithstanding, reduced ileal BA recycling was not compensated by an increase in the expression of BA transporters in the colon of CS-fed mice (Appendix A).

Once secreted into the portal blood system by ileocytes, BAs are reclaimed from the liver sinusoids via sodium-dependent and independent transporters. We observed a significant reduction in mRNA and protein expression of the main BA transporter at the sinusoidal side of hepatocytes (Figure 6A,C), the Na+-taurocholate cotransport protein (NTCP, SLC10A1), which is responsible for 80% of the clearance of conjugated BA from the portal vein [27,28,66]. Similarly to *Asbt, Ntcp* is down-regulated by FXR signaling [67,68,69]. However, FXR agonists/FXR antagonists ratio and FXR activation were significantly lower in the liver of CS-fed mice (Figure 2G and Figure 3B). We propose here that the rise in MCAs hinders FXR signaling in the liver. Therefore, it is unlikely that FXR signaling is involved in the regulation of NTCP expression in CS-fed mice. Because *Ntcp* and *Asbt* are HNF-1α target genes [69,70], we can speculate that NTCP is regulated by CS levels through a SREBP2/HNF-1α-dependent mechanism similar to ASBT [31]. Further investigations are required to address this hypothesis. Sodium-independent BA transport is quantitatively less important at the sinusoidal side of the hepatocytes. In rodents, it involves two members of the organic anions transporting protein family (OATP1 and OATP4; SLC21A1 and SLC21A10, respectively). In total, 80% of the Na+-independent BA influx depends on OATP1 [27,28,66]. Interestingly, mRNA levels of *Oatp1* were significantly reduced in the liver of CS-fed mice (Figure 6A).

In parallel, the expression of BA transporters involved in the efflux at the sinusoidal side of hepatocytes was unchanged with the exception of MRP4 (Figure 6B). These results are not surprising since this BA efflux system was reported to play a major role during cholestasis, which is characterized by a rise in sulfo- or glucurono-conjugated BAs that are excreted from the liver in the systemic blood circulation to be eliminated by the kidney [28,71,72,73]. In accordance with the reduced sinusoidal uptake of BAs in the liver, we noticed a moderate rise in plasma BAs levels in CS-fed mice (Figure 7A).

Circulating BAs are reclaimed by the epithelial cells of the kidney proximal convoluted tubules via a transport system similar to that of ileocytes [29]. Similar to the ileum, *Asbt* and *Ostα*/*β* expression was reduced upon dietary CS overload, while the expression of *Mrp2*, involved in apical BA efflux, as well as the expression of *Mrp3*, implicated in basal efflux of BAs in renal veins, were unchanged (Figure 7B–E). However, expression of *Mrp4*, which is also implicated in apical BA efflux, was dramatically upregulated in the kidney (Figure 7C). The resulting reduction in BA reabsorption in proximal convoluted tubules of the kidney was consistent with the enhancement of urinary BA excretion observed in CS-fed mice (Figure 7F).

Altogether, our data illustrate the onset of coordinated regulation of the key BA transporters in the gut, the liver, and the kidney in mice fed a CS diet, which contributes to increase BA outputs. This coordinated regulation might be driven by the blunting of FXR signaling by MCAs along with CS-mediated down-regulation of the SREBP-2 pathway. This coordinated regulation of the key BA transporters, especially NTCP in the liver and ASBT and OSTα/β in the gut and the kidney, results in a marked increase in BA output in feces and in urine, in favor of neo-synthesized BAs (Figure 10).

### 2.3. Fecal Elimination of Hydrophilic BAs Is a Major Route for Body CS Removal upon Dietary CS Overload

We propose that, in CS-fed mice, the enhancement of liver BA synthesis toward a more hydrophilic profile accompanied by the coordinated regulation of BA transporters in the liver, the gut, and the kidney favors BA elimination and contributes to body CS removal through conversion of CS into hydrophilic BAs. In agreement with this major contribution of BA in body CS removal, the analysis of publicly available data from the recombinant inbred strains of BXD mice (http://www.genenetwork.org/) revealed that BAs were among the top five metabolites in the feces (out of 3198 fecal metabolites identified by UPLC-MS2) of BXD mice fed a high fat diet (HFD) (data available from 9 and 11 strains for fecal Bas; data set: RTI RCMRC BXD Fecal Metabolites HFD Log2, GN Accession: GN716) with the strongest correlation with blood CS (data set: BXD_17802, 60% kCal/fat Harlan TD 06414, males, blood cholesterol mmol/L, EPFL LISP3 Cohort [74]) (Figure 8A). 

Figure 8B depicts the distribution of blood CS levels in 43 BXD strains fed either a standard chow diet (CD) (data set: BXD_17801, 6% kCal/fat Harlan 2918, males, blood cholesterol mmol/L, EPFL LISP3 Cohort [74]) or a HFD (data set: BXD_17802, 60% kCal/fat Harlan TD 06414), males, blood cholesterol mmol/L (EPFL LISP3 Cohort) [74]) and clearly shows the variation of cholesterolemia across BXD strains and diets. This further confirms the relevance BXD recombinant inbred strains to mimic the variability of biological parameters, as observed in human populations. Blood CS data and fecal metabolites (3198 fecal metabolites) were obtained from other research groups and only matched 7–12 BXD strains in terms of metabolites (9–11 strains for BA). The UPLC-MS2 broad spectrum metabolomics used did not allow us to discriminate between all BA species. We noticed a statistically significant negative correlation between tauroCDCA (TCDCA) or TDCA (*r* = −0.809, *p* = 2.559 × 10^−3^) as well as T-CA or T-MCA levels (*r* = −0.77, *p* = 9.22 × 10^−3^) in the feces and cholesterolemia in HFD-fed BXD strains (Figure 8A,C). Interestingly, we also noticed a statistically significant negative correlation between the variation in fecal TCA or TMCA levels and the variation in plasma CS levels in BXD strains fed a CD versus a HFD (Figure 8D). These data further confirmed that fecal elimination of hydrophilic BAs is a major route for body CS removal upon dietary CS overload. 

### 2.4. Fecal Cholesterol Elimination Is Increased in CS-Fed Mice

BA excretion represents a major physiological route for body CS removal. Nevertheless, it should not be disregarded that CS can also be eliminated via the biliary route [25,59]. At the canalicular side of hepatocytes, BAs and CS are excreted along with phospholipids (PL) to generate mixed micelles. The elimination of CS at the canalicular side of hepatocytes is mediated by the ABCG5/ABCG8 heterodimer [75]. MDR2, involved in phospholipids excretion, also plays a crucial role in this process since *Mdr2*-deficient mice fail to excrete CS in the bile even when ABCG5/G8 is overexpressed [76]. *Abcg5*, a well-established LXRα target gene, and *Mdr2* were up-regulated in the liver of CS-fed mice [77] (Figure 9A). 

**Figure 9 ijms-22-07163-f009:**
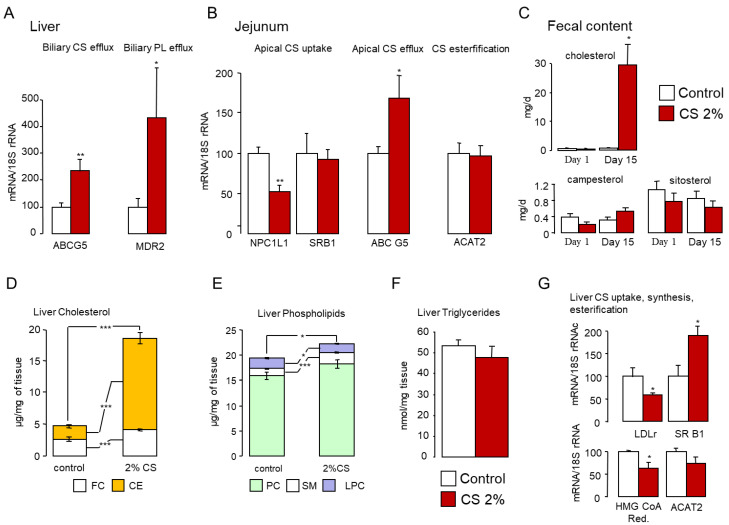
Fecal cholesterol excretion is increased in cholesterol-fed mice. (**A**,**B**) mRNA levels of sterol and phospholipids transporters in the liver (**A**) and the jejunum (**B**) of standard diet (control) or 2% cholesterol-fed (2% CS) mice. (**C**) Excretion of cholesterol and phytosterols in the feces of control and 2% cholesterol-fed mice. (**D**) Cholesterol, (**E**) phospholipid, (**F**) triglyceride levels in the liver of control and 2% cholesterol-fed mice. (**G**) mRNA levels of key lipoprotein receptors and cholesterol processing enzymes in the liver of control and 2% cholesterol-fed mice. Bar graphs represent mean ± s.e.m (*n* = 8 per group). * *p* < 0.05, ** *p* < 0.01, *** *p* < 0.001 determined by Student’s *t*-test with Welch’s correction.

These events were associated with a drop in *Npc1l1* expression in the jejunum where it acts as the main apical CS transporter in the gut [78]. In addition, *Abcg5* expression was enhanced (Figure 9B), suggestive of elevated apical CS efflux in enterocytes of the jejunum. SRB1, located in the microvilli of enterocytes, was initially proposed to contribute to CS uptake in the intestine [79]. Its expression was not affected by dietary CS overload, nor was expression of CS esterification enzyme ACAT2 (Figure 9B). Together, these gene expression changes account for the huge increase in CS output in the feces of CS-fed mice (Figure 9C) (2.99 ± 0.08 mg/d/100 g bw in control mice vs. 309.25 ± 6.18 mg/d/100 g bw in CS-fed mice). Of note, elimination of campesterol and sistosterol, two dietary phytosterols handled similarly to CS by intestinal transporters, was unaltered in CS-fed mice (Figure 9C), suggesting that the CS excreted in the feces is of biliary origin rather than from the diet. Notably, the BA hydrophobicity index (HI) is negatively correlated with fecal excretion of neutral sterol [80,81]. Decreased intestinal CS absorption triggered by hydrophilic BAs such as ursodeoxycholic acid (UDCA) or MCAs is namely due to reduced micellar solubilization of CS [82,83,84]. Therefore, in addition to the modulation of intestinal CS transporter expression, the high hydrophilic profile of the BA pool (enriched in MCAs) observed in CS-fed mice (Figure 2F and Figure 5F) also likely contributes to the high fecal CS release in these animals. 

Considering that food intake was not significantly altered in mice fed a diet containing 2% CS compared with mice on a standard diet (Appendix A), we can estimate that mice fed the CS-rich diet consumed approximately 120 times more CS than control mice (179.16 ± 30.63 mg/100 g of body weight versus 1.49 ± 0.29 mg/100 g of body weight, respectively). By contrast, liver CS content was multiplied by approximately 3.5 times, mainly in the form of cholesterol esters (Figure 9D). Liver phospholipid content was also slightly increased, whereas triglyceride levels were unaffected (Figure 9E,F). The uptake of CS in the liver was likely promoted by amplified expression of *Srb1,* while *Ldlr* expression declined, as did *Hmg-CoA reductase*, which drives endogenous CS synthesis in the liver (Figure 9G). This moderate increase in liver CS content compared to the major dietary CS overload highlights the remarkable propensity of mice to resist hypercholesterolemia. (A working model can be found as a summary in Figure 10).

**Figure 10 ijms-22-07163-f010:**
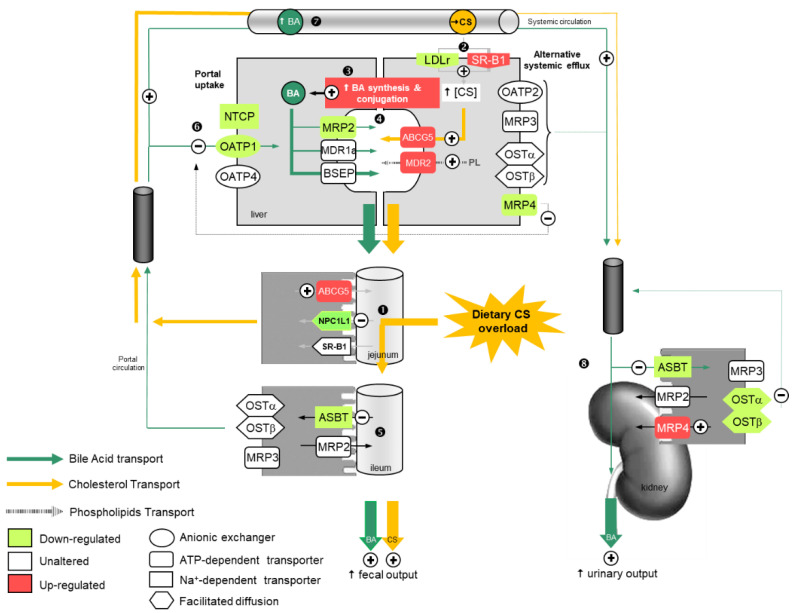
Working model of the coordinated regulation of bile acid and cholesterol elimination in cholesterol-fed mice. Dark green arrows relate to bile acid (BA) transport, yellow arrows to cholesterol (CS) transport, and gray arrows to phospholipid (PL) transport. Upregulated genes are depicted in red, unaltered genes are depicted in white, and downregulated genes in light green. ❶ Dietary CS from CS-rich diet is absorbed in the intestine. ❷ CS uptake is increased in the liver. ❸ Hydrophilic BA synthesis from CS and conjugation is augmented. ❹ Neo-synthetized BAs are secreted in bile ducts along with CS and PL. ❺ Expression of jejunal CS transporters and ileal BA recycling is diminished, resulting in enhanced elimination of neo-synthesized BAs and CS in feces. ❻ Reduction in portal reclamation of BAs results in ❼ elevation of blood BA levels. ❽ Reduced renal BA reclamation leads to increased urinary excretion.

## 3. Discussion

The mouse is a commonly used animal model for studying the physiopathology of dyslipidemias, notably hypercholesterolemia. Because mice are hypercholesterolemia-resistant [42,85], genetically engineered mouse models were generated to induce hypercholesterolemia (e.g., *Ldlr*^−/−^, *ApoE*^−/−^). The use of atherosclerotic diets containing a mixture of CS, BA, and saturated fatty acids is also traditionally employed to induce hypercholesterolemia in mice. Paradoxically, the mechanisms of resistance to hypercholesterolemia in mice are not fully elucidated. Understanding this process could provide new insights in the development of cholesterol lowering strategies and alert scientists with regard to specificities of CS metabolism in rodent models.

Cholesterol homeostasis and BA homeostasis are tightly linked. The latter is maintained by balancing hepatic synthesis of primary BA from CS, recycling by the liver via the enterohepatic cycle, and fecal and urinary elimination. In that, BA homeostasis relies on a tight regulation of liver BA synthesis pathways as well as BA transporters in the gut, the liver, and the kidney mediated by FXR, LXR, and SREBPs signaling [29,30,31,32]. Approximately 90% of CS is metabolized into BAs in the liver through two different pathways: the classical pathway and the alternative pathway. The classical pathway (CYP7A1) accounts for at least 75% of the total BAs, as demonstrated by the ablation of *Cyp7a1* in mice [33,34].

Intestinal absorption of lipids and lipophilic vitamins is severely impaired in *Cyp7a1*-deficient mice, resulting in increased perinatal mortality [33]. Interestingly, some mutations identified in the human *CYP7A1* gene are correlated with greater prevalence of hypercholesterolemia and atherosclerosis [86]. In line with this observation, *Cyp7a1*-deficient mice are hypercholesterolemic and display a proatherogenic phenotype [87]. The *Cyp7a1* gene is also differentially regulated in mice compared to humans. In mice and rats, unlike humans, *Cyp7a1* is upregulated by oxysterols upon CS feeding through LXRα activation [11,13,42,46]. Therefore, upon dietary CS overload, conversion of CS into BA is augmented through LXRα-mediated *Cyp7a1* upregulation. In the context of a Paigen diet, CA supplementation antagonizes CS conversion into BAs by repressing *Cyp7a1* expression in an FXR-dependent manner at the expense of LXR signaling. BA-mediated *Cyp7a1* down-regulation involves FXR-dependent upregulation of *SHP,* which subsequently binds to liver-receptor-homolog 1 (LRH-1) and LXRα, two master regulators of *Cyp7a1* gene expression. Moreover, in the ileum, FXR activation leads to *Fgf15* upregulation. FGF15 down-regulates *Cyp7a1* expression levels in the liver through FGFR4-βKlotho signaling [88]. The balance between FXR and LXR pathways controlling *Cyp7a1* gene expression remains controversial. The LXR pathway may be prevalent over the FXR pathway in mice and rats [13,46], whereas the opposite was documented in atherosclerosis-prone species such as rabbits or humans [48,49]. Upon CS overload, *Cyp7a1* expression decreases in the liver of rabbits and hamsters as well as in primary human hepatocytes [18,36,43]. 

Although BA metabolism plays a critical role in CS homeostasis and the onset of hypercholesterolemia, *Cyp7a1* deficiency is not always associated with reduced tissue CS turnover or hypercholesterolemia [89]. In this regard, our study not only focuses on the regulation of liver BA synthesis but also investigates the contribution of BA transporters in the liver, the gut, and the kidney to BA and CS homeostasis. We highlighted a coordinated regulation of BA transport in these tissues that is likely due to hydrophilic BA-mediated FXR deactivation. Rather than a dominance of LXR over FXR, our results suggest that the global BA hydrophobicity index (HI) is a key determinant of liver BA synthesis as well as BA transport in the liver, the gut, and the kidney. 

We propose that increased hepatic synthesis of hydrophilic BA in mice fed a CS-enriched diet associated with the coordinated regulation of BA transporters in the liver, the gut, and the kidney contributes to body CS removal through the conversion of CS into hydrophilic BA and their subsequent elimination in urine and feces along with biliary CS. Thus, our work highlights a central role for hydrophilic BAs in the resistance to hypercholesterolemia. This ability relies on their capacity to promote the LXR and the SREBP2 pathways at the expense of FXR signaling and to limit micellar solubilization of CS in the intestinal lumen, thereby favoring fecal elimination of CS [82,83,84]. This dual action of hydrophilic BA confers mice a striking ability to resist to hypercholesterolemia. Here, mice fed the CS-rich diet ingested 120 times the dose of dietary CS consumed by a mouse on a standard diet, yet cholesterolemia was unchanged, and liver CS content was only multiplied 3.5-fold. With less severe experimental settings, hypercholesterolemia-sensitive species (hamsters, rabbits) developed significant steatosis. Total liver CS levels rise from 2.5 mg/g in controls to 65 mg/g in 1% CS diet-fed hamsters (26-fold increase), while plasma CS increases 4.5-fold [15]. Similar results were also observed in CS-fed rabbit [20].

Our data suggest that inhibition of FXR signaling may be a relevant approach for the treatment of hypercholesterolemia. However, this avenue is controversial. Studies performed in *Fxr*^−/−^/*ApoE*^−/−^ or *Fxr*^−/−^/*Ldr*^−/−^ mice showed both protective and deleterious effects of FXR loss on atherogenesis and cholesterolemia [90,91,92,93]. The best pharmacological strategy to target hypercholesterolemia via FXR signaling is still questionable since the use of FXR agonists and antagonists generates interesting, albeit contentious results. FXR agonist GW4064 was shown to lower plasma cholesterol levels in mice, rats, and hamsters [94,95]. Strikingly, similar results were reported using Z-guggulsterone, which acts as an FXR antagonist [96]. Z-guggulsterone is the active component of guggulipid, a tree resin extract used in traditional Indian medicine to treat obesity and lipid disorders. Obeticholic acid (6-ethyl-CDCA) (OCA) is a potent FXR agonist approved by FDA for the treatment of primary biliary cholangitis and recently successfully passed a phase 3 trial for fibrosis due to NASH. Nevertheless, OCA treatment was associated with high LDL-cholesterol, triglycerides, and total cholesterol and reduced HDL-cholesterol [97], similar to what was previously described in hamsters treated with OCA or CDCA in patients with gallstones [98,99]. 

Our results suggest that the therapeutic potential of hydrophilic BAs relies on both FXR deactivation and their intrinsic plasma cholesterol lowering properties by physically reducing CS micellization and reabsorption in the gut lumen. Consistent with this hypothesis, the BA profile of *Cyp2c70*-deficient mice is rather hydrophobic due to the absence of MCAs and is associated with elevated plasma CS in both males and females, including those on a standard diet [39]. In humans, cholesterol-lowering properties of ursodeoxycholic acid, a common hydrophilic BA, were described in patients with primary biliary cirrhosis [100]. Therefore, the therapeutic potential of hydrophilic BAs for dyslipidemia and cardiovascular disorders should be emphasized in future approaches. In addition, BA composition greatly differs between hypercholesterolemia-resistant (mice, rats) and hypercholesterolemia-prone (NHP, humans, rabbits, hamsters) species. HI is higher in humans (0.45), hamsters (0.29), and rabbits (0.6) than in mice (−0.09) [41]. This discrepancy between human and mouse HI can be explained by the conversion of CDCA into MCAs by CYP2C70 [38,39], as high HI is negatively correlated with increase fecal excretion of neutral sterol [80,81]. The low HI of murine BA profiles also depends on the nature of the dominant BA species and the type of conjugation. Tri-hydroxylated (tri-OH) BAs are predominant in mice compared to mainly mono- and di-OH BAs in rabbits and humans. MCAs (α-, β-, and ω-MCA) (tri-OH, HI ≈ −0.6) and CA (tri-OH, HI ≈ +0.1) are the most abundant BAs in mice, whereas CDCA (di-OH, HI ≈ +0.6) and DCA (di-OH, HI ≈ +0.6) are the most represented BAs in humans, and DCA is the major BA in rabbits [41,81]. Tauro-conjugated BAs are mostly found in mice and are more hydrophilic than glyco-conjugated BAs that are more prevalent in humans and rabbits [29,101]. These distinct conjugation profiles are consistent with the idea that hypercholesterolemia-prone species such as rabbits and humans have a higher BA HI. 

To conclude, our study highlights an underestimated contribution of hydrophilic BAs and FXR deactivation in the resistance to hypercholesterolemia in mice. Further investigations are required to assess the therapeutic potential of this subset of BAs in the treatment of hypercholesterolemia. Additionally, species-specific CS and BA metabolism should lead scientists to caution when translating mouse data to human applications in the field of dyslipidemia and cardiovascular disorders.

## 4. Materials and Methods

### 4.1. Animal Studies

All studies with mice were conducted in accordance with the local guidelines for animal experimentation. Protocol no. 2512 was approved by the institutional animal care and use committee of the Université de Bourgogne-Franche-Comté. The 8-week-old male C57BL6/J mice purchased from Charles River (France) were housed in individual metabolic cages after a 5-day acclimation period in a temperature-controlled room and were maintained in a light/dark cycle. Mice were fed either a standard chow (UAR A04, 0.02% CS, *w*/*w*) (control) or a standard chow supplemented with 2% cholesterol (*w*/*w*) (UPAE, INRA, Jouy-en-Josas) (2% CS-fed mice) for 15 days. Feces and urine were collected every two days. Body mass as well as food and water consumption were assessed every day. Firefly luciferase reporter mice for the nuclear receptor farnesoid-X-receptor (*Fxr*-Luc reporter mice [54]) were housed in a temperature- and humidity-controlled facility and fed a standard chow diet or a 2% CS diet for 15 days. Experiments were performed with age-matched male mice (8 weeks old) as previously reported [54,102].

At the end of the diet period, animals were fasted overnight and euthanized under isoflurane anesthesia. Plasma was recovered from blood by centrifugation, ileal mucosa was scraped, and the liver and the kidney were sectioned in small pieces. All samples were immediately snap frozen in liquid nitrogen and stored at −80 °C until use.

### 4.2. Quantification of Fecal Bile Acids and Sterols

Feces were dried immediately following collection and crushed into a fine powder. Nor-cholic acid and cholestane were used as internal standards. The procedure of derivatization was adapted from Batta et al. [102]. Analysis was performed by gas chromatography-mass spectrometry (Agilent Technologies, Les Ulis, France).

### 4.3. Quantification of Liver Lipids and Bile Acids and Liver Enzymatic Activities

Liver samples were homogenized in 150 mM NaCl. Bile acid analysis: The procedure of sample extraction was adapted from Keller et al. [103]. Nor-cholic acid was used as internal standard. Analysis was performed by gas chromatography-mass spectrometry. Cholesterol analysis: Epicoprostanol was used as internal standard. Analysis was performed by gas chromatography-mass spectrometry [104]. Phospholipids analysis: Phosphatidylcholine,1,2-dimyristoyl was used as internal standard. Analysis was performed by quantitative liquid chromatography-mass spectrometry. Analysis with a Hypersil Si 2x200 mm column in a HPLC 1100 (Agilent Technologies) [105]. Triglycerides analysis: Hepatic TGs were quantified using the Triglycerides 25 kit according to the manufacturer’s instructions (ABX Diagnostics, Montpellier, France). Enzymatic activities of cyp7a1 and cyp8b1 were assessed as previously reported [106].

### 4.4. Quantification of Plasma Lipids and Bile Acids

Plasma total cholesterol (Cholesterol 100, ABX diagnostics), free CS (Free Cholesterol C-R1, Wako), and plasma TG (triglycerides 25, ABX Diagnostics) concentrations were determined by enzymatic assays according to the manufacturer’s instructions. Plasma and urinary BA were also quantified with an enzymatic kit (Colorimetric Total Bile Acid Assay Kit, BioQuant).

### 4.5. Western Blotting

A total of 15 µg of total proteins extracted from the ileal mucosa, the liver, or the kidney were prepared in ice-cold buffer (0.154 M KCl, 0.01 M phosphate buffer, pH = 7.4), separated on 10% SDS-PAGE, then blotted onto a Polyscreen membrane (PerkinElmer Life Sciences). Rabbit polyclonal anti-ASBT, NTCP, BSEP, MRP2, OSTα, and OSTβ were a generous gift from Dr. M. Souidi (Institut de Radioprotection, Fontenay-aux-Roses, France), Prof. M. Trauner (Gastroenterology and Hepatology, Vienna, Austria), Prof. N. Ballatori University of Rochester School of Medicine, Rochester, NY, USA), and Prof. G. Kullak-Ublick (University Hospital Zurich, Zurich, Switzerland). Anti-b-actin antibodies and anti-rabbit peroxidase-conjugated secondary antibodies were purchased from Sigma-Aldrich. ECL blotting kit (PerkinElmer Life Sciences) was used for detection. Blot quantification was performed using calibrated GS-800 densitometer (Biorad, Marnes-la-coquette, France).

### 4.6. Quantitative Real-Time PCR

cDNA was produced from 1 µg total RNA by reverse transcription (Omniscript Reverse Transcription, Qiagen) after a preliminary DNase treatment (DNase I, amplification Grade, Invitrogen life technologies). The amount of cDNA synthesized from 25 ng total RNA was mixed with qPCR Mastermix Plus for SYBR Green I Fluorescein (Eurogentec) and 200 nM of primers. For all primers used for mRNA quantification, PCR accuracy and efficiency were previously assessed. Primer sequences are available upon request. PCR consisted of a denaturation at 95 °C for 15 s followed by annealing at 60 °C for 30 s (except for Srb1, 56 °C for 30 s). Real-time PCR was performed using the ICycler IQ (Biorad) detection system. Quantification was based on the comparative ΔΔCt method. mRNA levels were normalized to 18S rRNA.

### 4.7. Statistical Analysis

The results are expressed as means ± s.e.m. All experimental values were obtained from the measurement of distinct samples and non-repeated measures of the same sample. The number of replicates is indicated in the Figure legends. Statistical differences were analyzed with GraphPad Prism 8. Comparison between groups was performed using unpaired Student’s *t*-test with Welch’s correction. Correlations were analyzed using Spearman’s rank correlation. A *p* value of less than 0.05 was considered statistically significant. Sample sizes were selected on the basis of previous knowledge of the variation in experimental methods and the expected effect size observed in previous studies. Variance was similar between groups compared and followed normal distribution.

## Figures and Tables

**Figure 1 ijms-22-07163-f001:**
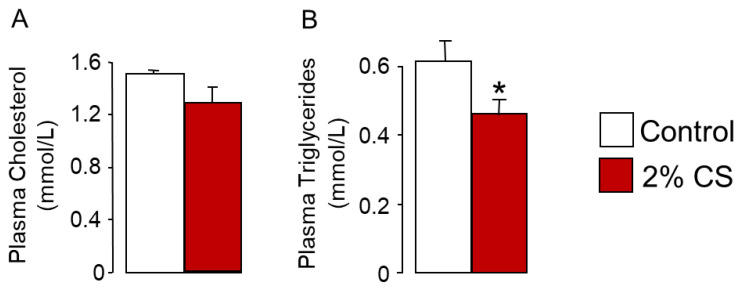
Mice fed a cholesterol-enriched diet did not develop dyslipidemia. (**A**) Plasma cholesterol and (**B**) plasma triglycerides levels in 8-week-old mice fed a standard diet (control) or a 2% cholesterol-enriched diet for 15 days (2% CS). Bar graphs represent mean ± s.e.m (*n* = 8 per group). * *p* < 0.05, determined by Student’s *t*-test with Welch’s correction.

**Figure 2 ijms-22-07163-f002:**
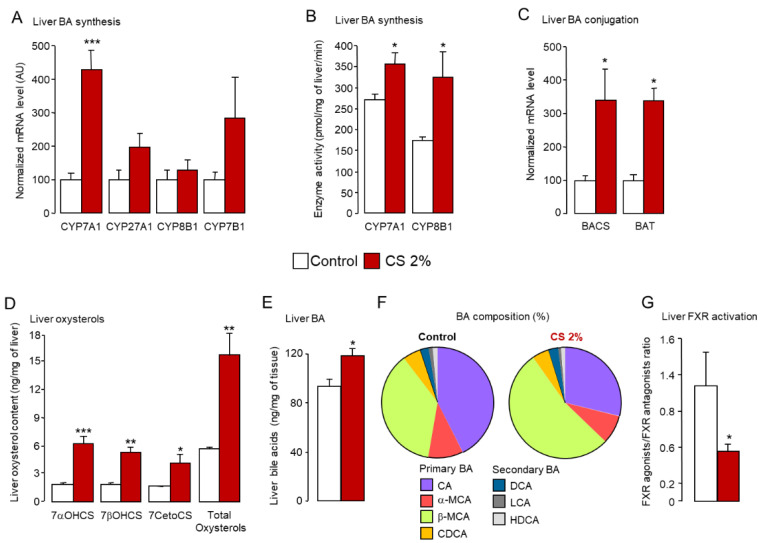
Liver synthesis of hydrophilic bile acids is markedly enhanced in cholesterol-fed mice. (**A**–**C**) mRNA level and enzymatic activity of key enzymes of bile acid synthesis and conjugation in the liver of standard diet (control) or 2% cholesterol-fed (2% CS) mice. (**D**,**E**) Oxysterols and bile acid levels in the liver of standard diet (control) or 2% cholesterol-fed (2% CS) mice. (**F**) Liver bile acid composition in standard diet (control) or 2% cholesterol-fed (2% CS) mice. (**G**) Ratio of FXR agonists (CA, CDCA, DCA) and FXR antagonists (αMCA, βMCA, UDCA) in the liver of standard diet (control) or 2% cholesterol-fed (2% CS) mice. Bar graphs represent mean ± s.e.m (*n* = 8 per group). * *p* < 0.05, ** *p* < 0.01, *** *p* < 0.001 determined by Student’s *t*-test with Welch’s correction.

**Figure 3 ijms-22-07163-f003:**
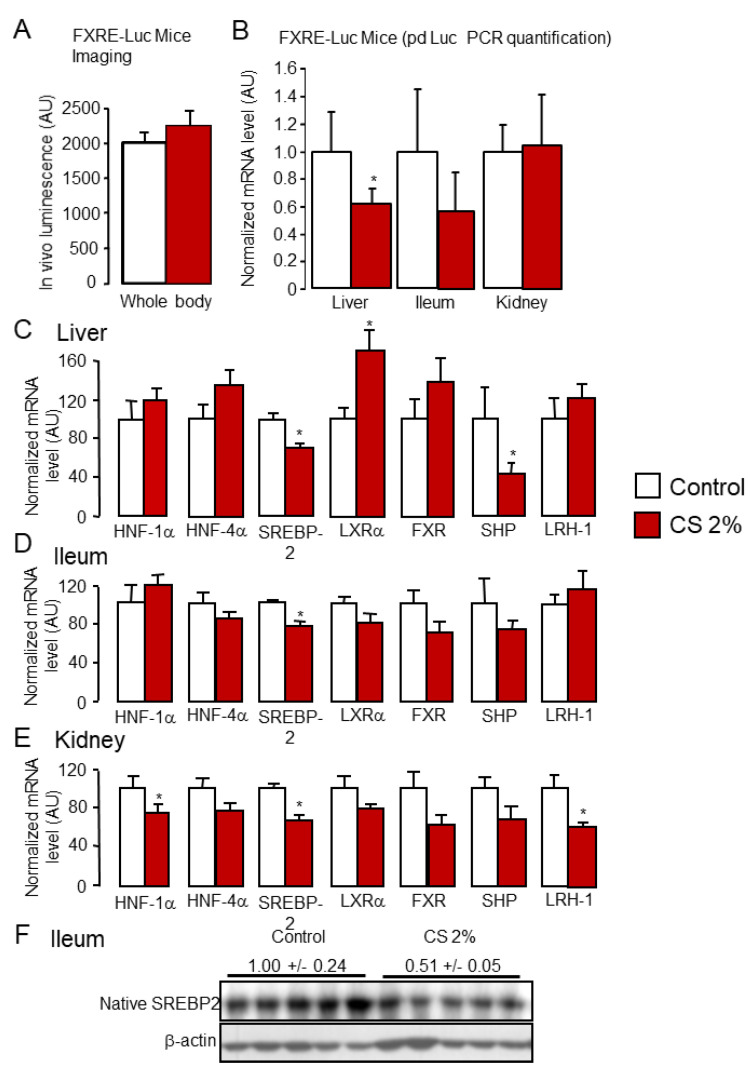
FXR signaling is reduced in cholesterol-fed mice. (**A**) Total body imaging of FXR-driven luciferase levels in standard diet (control) or 2% cholesterol-fed (2% CS) FXRE-luc mice. (**B**) mRNA levels of pd-Luc (luciferase encoding mRNA) in the liver, the ileum, and the kidney of standard diet (control) or 2% cholesterol-fed (2% CS) FXRE-luc mice. (**C**–**E**) mRNA levels of key transcription factors and nuclear receptors regulating bile acids and cholesterol homeostasis in the liver (**C**), the ileum (**D**), and the kidney (**E**) of standard diet (control) or 2% cholesterol-fed (2% CS) mice. (**F**) Protein levels of mature form of SREBP2 transcription factor in the ileum of standard diet (control) or 2% cholesterol-fed (2% CS) mice. * *p* < 0.05.

**Figure 4 ijms-22-07163-f004:**
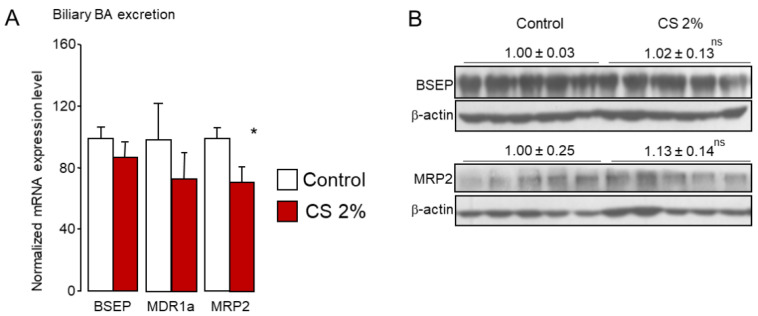
Liver canalicular excretion of bile acids is unchanged in cholesterol-fed mice. (**A**) mRNA and (**B**) protein levels of canalicular bile acid transporters in the liver of standard diet (control) or 2% cholesterol-fed (2% CS) mice. Bar graphs represent mean ± s.e.m (*n* = 8 per group). * *p* < 0.05 determined by Student’s *t*-test with Welch’s correction.

**Figure 5 ijms-22-07163-f005:**
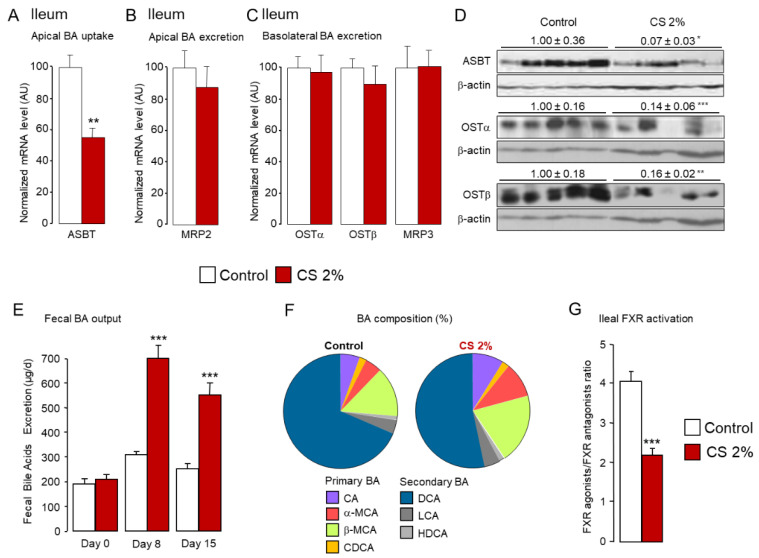
Reduced ileal bile acid recycling leads to increased fecal outputs of bile acids in cholesterol-fed mice. (**A**–**C**) mRNA and (**D**) protein levels of bile acid transporters in the ileum of standard diet (control) or 2% cholesterol-fed (2% CS) mice. (**E**) Excretion of bile acids and (**F**) composition of the bile acid pool in the feces of standard diet (control) or 2% cholesterol-fed (2% CS) mice. (**G**) Ratio of FXR agonists (CA, CDCA, DCA) and FXR antagonists (αMCA, βMCA, UDCA) in the feces of standard diet (control) or 2% cholesterol-fed (2% CS) mice. Bar graphs represent mean ± s.e.m (*n* = 8 per group). * *p* < 0.05, ** *p* < 0.01, *** *p* < 0.001 determined by Student’s *t*-test with Welch’s correction.

**Figure 6 ijms-22-07163-f006:**
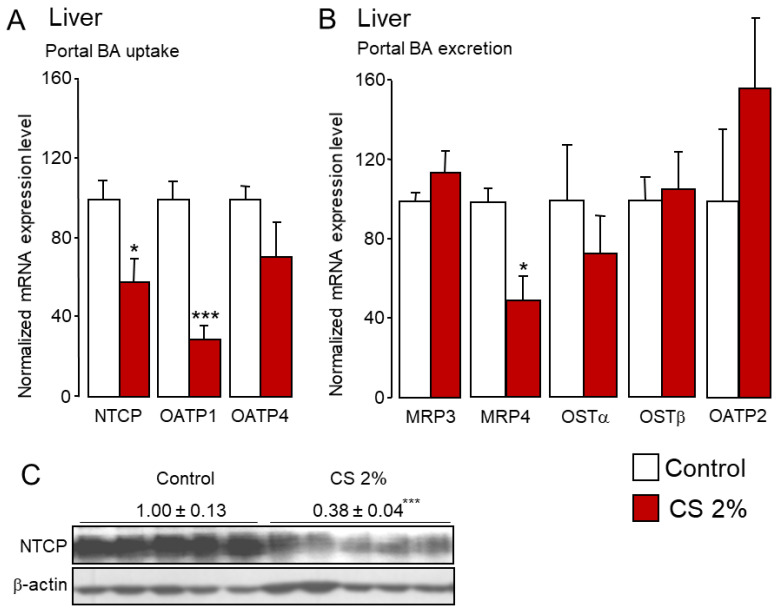
Liver sinusoidal reclamation of bile acids is reduced in cholesterol-fed mice. (**A**,**B**) mRNA and (**C**) protein levels of sinusoidal bile acid transporters in the liver of control and 2% cholesterol-fed mice. Bar graphs represent mean ± s.e.m (*n* = 8 per group). * *p* < 0.05, *** *p* < 0.001 determined by Student’s *t*-test with Welch’s correction.

**Figure 7 ijms-22-07163-f007:**
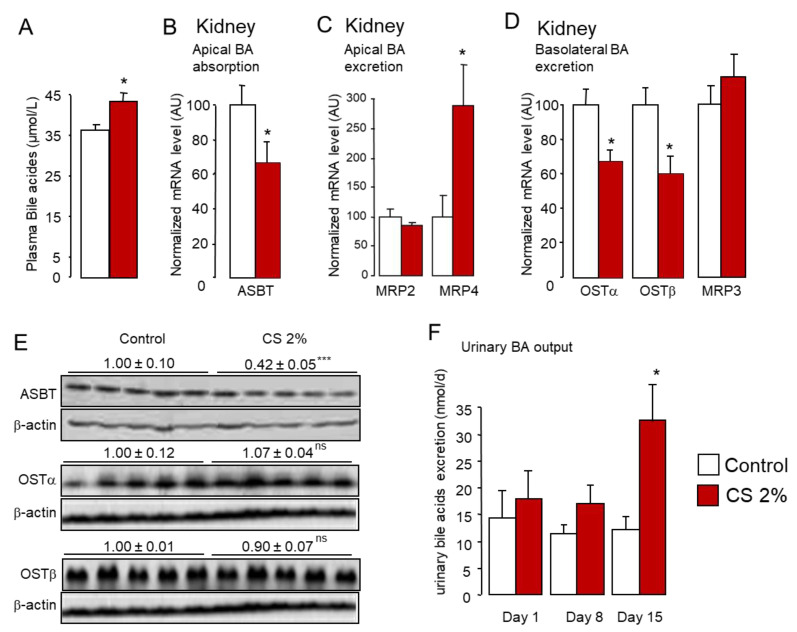
Reduced renal bile acid reclamation leads to increased urinary excretion of bile acids in cholesterol-fed mice. (**A**) Plasma levels of bile acids in standard diet (control) or 2% cholesterol-fed (2% CS) mice. (**B**–**D**) mRNA and (**E**) protein levels of bile acid transporters in the kidney of standard diet (control) or 2% cholesterol-fed (2% CS) mice. (**F**) Excretion of bile acids in urine of standard diet (control) or 2% cholesterol-fed (2% CS) mice. Bar graphs represent mean ± s.e.m (*n* = 8 per group). * *p* < 0.05, *** *p* < 0.001 determined by Student’s *t*-test with Welch’s correction.

**Figure 8 ijms-22-07163-f008:**
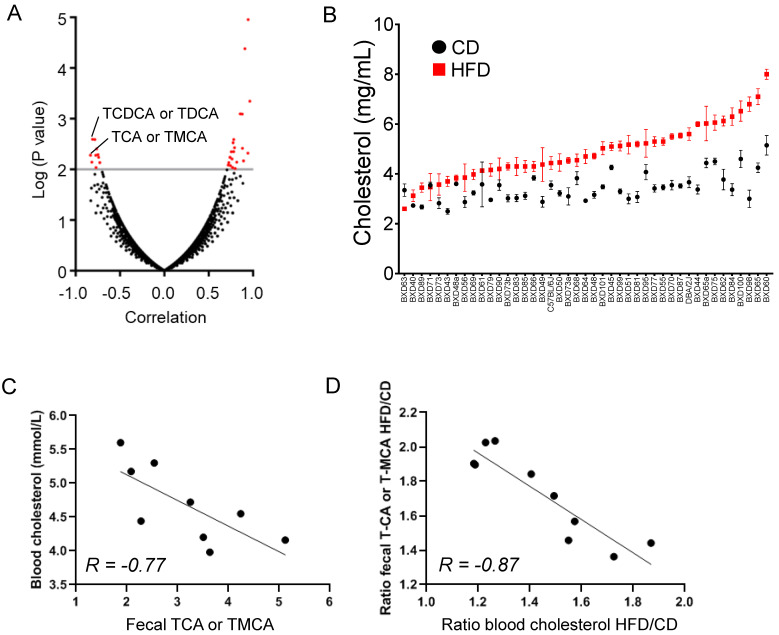
Increased fecal bile acid excretion is correlated with lowered cholesterolemia in high fat diet-fed BXD mice. (**A**) Volcano plot of the spearman correlation rank of fecal metabolites and plasma cholesterol level in BXD recombinant inbred mice using publicly available data from Genenetwork (http://www.genenetwork.org/). (**B**) Distribution of plasma cholesterol levels in standard diet and high fat diet fed in 43 BXD recombinant inbred strains. (**C**) Spearman correlation rank of fecal bile acid (taurocholic acid (TCA) or tauromuricholic acid (TMCA)) levels and plasma cholesterol levels in high fat diet fed 9 BXD recombinant inbred strains. (**D**) Spearman correlation rank of fecal bile acid (ratio of taurocholic acid (TCA) or tauromuricholic acid (TMCA) in chow diet fed to high fat diet-fed BXD recombinant inbred mice) and variation (ratio) of plasma cholesterol levels in the same BXD recombinant inbred strains fed a chow diet or a high fat diet.

## Data Availability

No data suitable for public databases were obtained.

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
