# Peer review of "Muricholic Acids Promote Resistance to Hypercholesterolemia in Cholesterol-Fed Mice"

_ijms, 2021, doi:10.3390/ijms22137163_

Round 1
Reviewer 1 Report
Based on a solid rationale, the Authors hypothesized that hepatic conversion of cholesterol into bile acids participates in mice's resistance to diet-induced hypercholesterolemia. Through well connected and significant dataset, the Authors highlight the contribution of hydrophilic bile acids and FXR signaling deactivation in the resistance to hypercholesterolemia. All these coordinated events are mediated by increased muricholic acid levels, which inhibit FXR signaling in favor of LXR and SREBP2 signaling to promote efficient fecal and urinary elimination of cholesterol and bile acids. The related regulation of bile acids transporters in the liver, gut, and kidney are exhaustively evaluated and discussed. This is outstanding work.
Minor point
Please, check figure numbers that do not correspond: line 171, 216, 240, 244, 366.
Biles acids HI index should be calculated and shown on figures or tables.
Line 57: please remove "only"
Line 236: please add percentage proportion to the following sentence: "Whereas the proportion of primary BAs in the feces was enhanced, the % of fecal excretion of secondary BA deoxycholic acid (DCA), produced from CA by colonic bacteria, was significantly reduced (Figure 5F and Table S2)."
While the proportion of fecal DCA is decreased, the overall fecal excretion of DCA is increased, and the ratio Primary BA/secondary BA is doubled (0.35 versus 0.68) in cholesterol-fed mice. Can this phenomenon impact directly or indirectly cholesterolemia?
Primary BA/ Secondary BA ratios should be indicated in Tables and potentially discussed.
Author Response
Reviewer 1:
Based on a solid rationale, the Authors hypothesized that hepatic conversion of cholesterol into bile acids participates in mice's resistance to diet-induced hypercholesterolemia. Through well connected and significant dataset, the Authors highlight the contribution of hydrophilic bile acids and FXR signaling deactivation in the resistance to hypercholesterolemia. All these coordinated events are mediated by increased muricholic acid levels, which inhibit FXR signaling in favor of LXR and SREBP2 signaling to promote efficient fecal and urinary elimination of cholesterol and bile acids. The related regulation of bile acids transporters in the liver, gut, and kidney are exhaustively evaluated and discussed. This is outstanding work.
We thank reviewer 1 for his/her positive and helpful comments.
Minor point
Please, check figure numbers that do not correspond: line 171, 216, 240, 244, 366.
We checked and corrected all figure numbers in the ms.
Biles acids HI index should be calculated and shown on figures or tables.
We thank reviewer 1 for this comment. As suggested, the HI values have been added in Tables S1 and S2 as reported in the literature (Heuman, D.M. Quantitative Estimation of the Hydrophilic-Hydrophobic Balance of Mixed Bile Salt Solutions. J. Lipid Res. 1989, 30, 719–730). Dedicated HPLC experiments are required to determine the bile acid pool HI index of each mouse. Unfortunately, HI of the bile acid pool of each mouse cannot be calculated with the data obtained in the context of this work (GC-MS data). Although of interest, this parameter can be properly estimated with the data from the literature as now presented in Tables S1 and S2.
Line 57: please remove "only"
The correction has been done.
Line 236: please add percentage proportion to the following sentence: "Whereas the proportion of primary BAs in the feces was enhanced, the % of fecal excretion of secondary BA deoxycholic acid (DCA), produced from CA by colonic bacteria, was significantly reduced (Figure 5F and Table S2)."
The sentence has been updated as follow: “Whereas the proportion of primary BAs in the feces was enhanced (26.4±1.25% in control vs 40.7±3% in 2% CS fed mice, p=0.001), the fecal excretion of secondary BA deoxycholic acid (DCA), produced from CA by colonic bacteria, was significantly reduced (68.6±1.2% in control vs 53.3±2.6% in 2% CS fed mice, p=0.0003) (Figure 5F and Table S2).”
While the proportion of fecal DCA is decreased, the overall fecal excretion of DCA is increased, and the ratio Primary BA/secondary BA is doubled (0.35 versus 0.68) in cholesterol-fed mice. Can this phenomenon impact directly or indirectly cholesterolemia?
We thank reviewer 1 for this question. When considering the amount of excreted BAs, the excretion of both primary and secondary BAs is increased (see Table S2). Since, the total amount of primary BA is greatly enhanced in CS-fed mice, when expressed in percentage, it appears that the proportion of secondary BA excreted (and hence DCA) is decreased. Therefore, when expressed in percentage, secondary BAs are proportionally less excreted than primary BA. The excretion is in favor of primary BAs which are also more abundant in CS-fed mice. Indeed, this will impact on cholesterolemia since the subsequent enhancement of BA synthesis (primary BAs) will “use” liver CS and will therefore avoid hypercholesterolemia in CS-fed mice. This point is discussed in several sections of our article, e.g line 370: “We propose that in CS-fed mice, the enhancement of liver BA synthesis toward a more hydrophilic profile accompanied by the coordinated regulation of BA transporters in the liver, the gut and the kidney, favors BA elimination and contributes to body CS removal through conversion of CS into hydrophilic BAs” or line 542: “We propose that, increased hepatic synthesis of hydrophilic BA in mice fed a CS-enriched diet, associated with the coordinated regulation of BA transporters in the liver, the gut and the kidney, contributes to body CS removal through the conversion of CS into hydrophilic BA and their subsequent elimination in urine and feces along with biliary CS. Thus, our work highlights a central role for hydrophilic BAs in the resistance to hypercholesterolemia.“
Primary BA/ Secondary BA ratios should be indicated in Tables and potentially discussed.
We thank reviewer 1 for this comment. These ratios have been added in Table S1 and S2.

Reviewer 2 Report
The study highlight some of the limitatioons linked to animl models of NAFLD. The major limitation of the study is that there is no mechanistic evidence that MCA are the only bile acids that explain the different impact of the cholesterol diet in mice vs humans.
Data on Tauro-conjugated bile acids should be shown in the addition to free bile acids
Author Response
he study highlight some of the limitatioons linked to animl models of NAFLD. The major limitation of the study is that there is no mechanistic evidence that MCA are the only bile acids that explain the different impact of the cholesterol diet in mice vs humans.
We thank reviewer 2 for his/her comments. We understand the comment of reviewer 2. We reviewed in the introduction section the mechanisms described in the literature that contribute to explain the differences in term of sensitivity to hypercholesterolemia among several species (namely mice, rat, hamster, rabbit and human). Based on that, our work proposed and showed that the hydrophilic profile of BA pool in mice (mainly driven by the high amount of MCAs) contribute to explain the striking ability of the mouse to resist to hypercholesterolemia upon CS feeding. Mechanistically, we showed that the enhancement of MCA levels leads to FXR deactivation. We used a comprehensive approach that would be very complex to perform in human subjects. Moreover, we want to point out that MCAs are absent sin humans. Therefore, supplementation with MCA in humans would be required. We agree that this would be interesting to carry out such intervention approaches in patients in the future. This is beyond the scope of this current work.
We used BXD recombinant inbred mice to mimic the genetic diversity observed in human populations. Our data showed that there is a strong correlation between the amount of hydrophilic BAs in the feces and the resistance to hypercholesterolemia among mouse BXD strains.
Data on Tauro-conjugated bile acids should be shown in the addition to free bile acids
We thank reviewer 2 for this comment. BA analysis were performed by GC-MS. This does not allow measuring conjugated BAs. We believe that this does not represent a major issue since approximately 98% of BAs are conjugated in mice and mostly to taurine. Therefore, the BA profile depicted in our study is very similar to this of tauro-conjugated BAs.
